# Genetic buffering and potentiation in metabolism

**Juan F. Poyatos**[1,2]*

**1** Logic of Genomic Systems Lab (CNB-CSIC), Madrid, Spain, **2** Center for Genomics and Systems Biology (NYU), New York, United States of America

* jpoyatos@cnb.csic.es

**Data Availability Statement:** All relevant data are within the manuscript and its Supporting Information files.

**Funding:** This work was supported in part through the NYU IT High Performance Computing resources, services, and staff expertise, grants

## Abstract

Cells adjust their metabolism in response to mutations, but how this reprogramming depends on the genetic context is not well known. Specifically, the absence of individual enzymes can affect reprogramming, and thus the impact of mutations in cell growth. Here, we examine this issue with an *in silico* model of *Saccharomyces cerevisiae's* metabolism. By quantifying the variability in the growth rate of 10000 different mutant metabolisms that accumulated changes in their reaction fluxes, in the presence, or absence, of a specific enzyme, we distinguish a subset of modifier genes serving as buffers or potentiators of variability. We notice that the most potent modifiers refer to the glycolysis pathway and that, more broadly, they show strong pleiotropy and epistasis. Moreover, the evidence that this subset depends on the specific growing condition strengthens its systemic underpinning, a feature only observed before in a toy model of a gene-regulatory network. Some of these enzymes also modulate the effect that biochemical noise and environmental fluctuations produce in growth. Thus, the reorganization of metabolism induced by mutations has not only direct physiological implications but also transforms the influence that other mutations have on growth. This is a general result with implications in the development of cancer therapies based on metabolic inhibitors.

## Author summary

Identical genetic changes do not always lead to the same phenotype and can thus contribute in different ways to phenotypic variation. These context-dependent effects are usually associated with the presence, or absence, of elements identified as genetic modifiers. Highly specific proteins with global action in the cell, like the molecular chaperone Hsp90, were initially recognized as modifiers. Later work showed that this context dependence is a general characteristic of molecular networks. This was demonstrated with a toy model of a gene-regulatory network. Here, we use genome-scale metabolic network modeling to examine for the first time the latent function of enzymes as modifiers that can suppress (buffer) or amplify (potentiators) the impact of mutations in the phenotype (in this case growth rate). Our results emphasize how context dependence is an intrinsic feature of the system generating the phenotype rather than of its constituents. We also

FIS2016-78781-R and the Salvador de Madariaga
program (grant PRX18/00439) from the Spanish
Ministerio de Economía y Competitividad, and
grant PID2019-106116RB-I00 from the Spanish
Ministerio de Ciencia e Innovación. The funders
had no role in study design, data collection and
analysis, decision to publish, or preparation of the
manuscript.

**Competing interests:** The authors have declared
that no competing interests exist.

discuss the implications of this analysis for our understanding of the consequences of metabolic reprogramming in cancer progression.

## Introduction

Cells experience mutations in different ways. The direct importance of these on the phenotype has been the focus of substantial basic and applied research [1, 2]. It is much less known, however, how specific genetic contexts modify the phenotypic impact of mutations [3, 4], and the many consequences that the alterations could have on disease progression [5].

One can expect two broad situations. In the first one, the presence of particular genetic variants buffers the effect of mutations. This result helps explain the robustness observed in biological phenotypes and was already discussed–under the notion of canalization–in early studies of development [6–8]. Canalization, or robustness, also leads to the accumulation of cryptic genetic variation [9, 10], which does not reveal under typical conditions. Therefore, the unveiling of this hidden variation after perturbation was reported as a decline of robustness. However, this is not necessarily so [11, 12]: two systems presenting the same robustness can nevertheless expose cryptic variation linked to mutations which are neutral depending on the system they emerge (conditional neutrality) [10–12]. Moreover, a second general scenario corresponds to the case in which some genetic variants potentiate the functional consequences of mutations what can eventually promote the rapid evolution of new traits [13, 14].

This wide range of implications encouraged the search for the genetic underpinnings of buffering or potentiation. And thus, the chaperone Hsp90 was the first described protein deemed to be a canalization agent whose altered function leads to more significant phenotypic variation, a result initially demonstrated in *Drosophila* [15] and later generalized across species [16–18]. Hsp90 represents in this way a buffer or capacitor (because its influence resembles the storage and subsequent release of electrical charge by a *capacitor* in electrical circuits). Indeed, its consequences on the folding and stability of other proteins fit well with the notion of a global element contributing to the canalized phenotype, a role also attributed to a few additional molecular agents, like the prion [PSI$^+$] [19].

But later studies raised some doubts on the action, definition, and uniqueness of particular proteins as capacitors. For instance, in the precise case of heat shock proteins, part of the associated variation is linked to their control of the mutagenic activity of transposons [20]. Besides, these proteins can not only reduce but also amplify the impact of mutations by making them produce immediate phenotypic consequences. The same molecular element is then modifying the impact of mutations in two contrasting ways [13, 14]. Other uncertainties indicate constraints on the conventional experimental approach to examine these issues, in which selection sieves the mutations commonly assayed. Mutation accumulation experiments [21] reduce the strength of selection and thus provide a more accurate sample instead [22].

Beyond these objections, a more important criticism is the evidence that buffers, or potentiators, are not fundamentally connected to special molecular agents with distinct biochemical properties but that they emerge as an intrinsic feature of complex biological networks. Many genes could accordingly modify the effect of mutations [23]; a prominent conclusion if one were to bring in the earlier results as part of the representative methodology of genetics [1] but maybe less unexpected in the broader framework of the architecture of complexity [24].

The main focus of this manuscript is to consider metabolism as a representative model system to examine whether buffering and potentiation is indeed a common phenomenon in biological networks. While this result has been shown with the use of toy gene-regulatory

networks [23], and the finding of new genes acting as capacitors appears to confirm this conclusion [12], its validation in realistic biological networks is still lacking. Moreover, and given that the experimental manipulations accompanying this question are challenging, we contemplate instead an *in silico* representation of metabolism. These genome-scale metabolic models have become a standard in systems biology, contain all of the known metabolic reactions in an organism, and the genes encoding each enzyme. They can be used to compute the flow of metabolites through the metabolic network and predict the growth rate [25]. Earlier work on robustness and evolution of metabolic networks confirms the soundness of this approach [26–28].

We thus consider a genome-scale reconstruction of *Saccharomyces cerevisiae* [29] to examine if the architecture of metabolic networks facilitates buffers and potentiators. To this aim, we examine to what extent the presence of a particular enzyme changes the influence on the growth rate of a compendium of mutations altering the metabolic fluxes. We thus generate a collection of mutant metabolisms (mutation accumulation lines) derived from the wild-type, which displays a well-defined variability in the growth rate. We then quantify if these *very same lines* manifest a different variability depending on the absence of a single enzyme. This led us to identify a set of genes acting as buffers and potentiators whose influence depends on the particular working conditions of the metabolism (i.e., type of available nutrients), and the sources of variability considered. Therefore, buffering and potentiation do not only depend on the structure of the metabolic network but also on its mode of operation. We finish evaluating how this fundamental phenomenon could have practical implications in the development of metabolic-based cancer therapies and the wide-ranging use of modifiers genes to control disease.

## Results

### Buffers and potentiators in metabolism

We examined the significance of each metabolic enzyme on how mutations impact the growth rate, which is regarded here as a case study of a complex phenotype. To this aim, we generated a collection of mutant metabolisms simulating the production of spontaneous mutations in independent cell lines, like those obtained in mutation accumulation (MA) experiments [21]. These kinds of collections help characterize the response of biological systems to new mutations that did not experience any purge by selection [11].

In this metabolic setting, we first derived the mutant compendium by limiting the flux of 5% of the total reactions chosen randomly in the wild-type metabolism (Fig 1A). We obtained in this way 10000 different mutant lines, a feasible number to generate *in silico*, but a challenging one to reproduce experimentally (a typical MA collection contains about 100 lines [21]). For each member of the compendium, we compute its growth rate ("fitness") by minimizing the metabolic adjustment caused by the mutations on the fluxes of the wild-type metabolism, an approach that is known to successfully predict growth rates and fluxes upon mutation [30]. Each line included in the collection presents nonzero fitness (see Methods for details).

We then computed the relative effect of the former MA lines in any metabolism in which an individual enzyme has been deleted, i.e., the mutations constituting the MA lines are fixed (Fig 1B). The difference in phenotypic (growth-rate) variation in the presence or absence of an enzyme reveals how it modifies the consequences of flux mutations on fitness. We quantified this difference with a score defined by the change between standard deviations $\theta = (\text{std}_{\text{mutant}} - \text{std}_{\text{wild-type}})/\text{std}_{\text{wild-type}}$, with $\theta < 0$ indicating that the enzyme works as a potentiator (presence of the enzyme increases variability) and $\theta > 0$ indicating that it acts as a buffer [presence of the enzyme decreases variability, Methods [11, 22]].

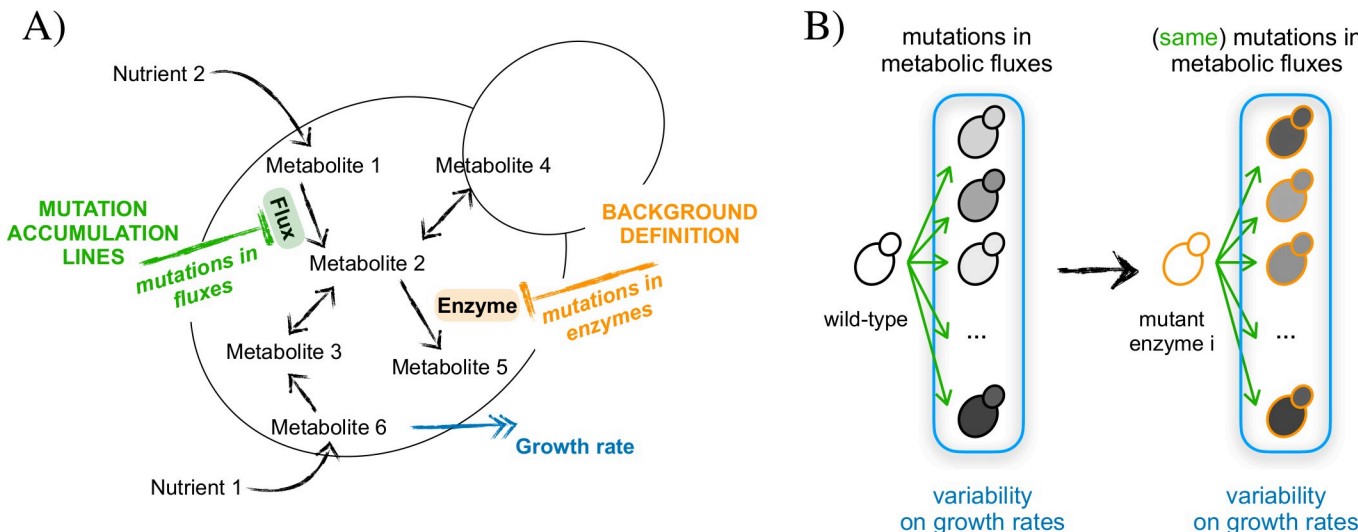

**Fig 1. Influence of global modifiers in metabolism.** A) An *in silico* representation of yeast metabolism can experience two types of mutations: 1)mutations in metabolic fluxes, which define the mutation accumulation lines, and 2)mutations in the enzymes, which define the particular backgrounds. The complex phenotype considered is growth rate (relative to the corresponding growth rate of each reference metabolism). B) We score the variability of the growth rates in a group of different lines (arrows) in which mutations in the metabolic fluxes are accumulated. We compute this variability in the presence (wild-type) and the absence (mutant metabolism) of a particular enzyme i. Here the difference between growth rates and metabolic backgrounds is represented by the colors of the fill and the border of the yeast cartoons, respectively.

Under a nutrient-rich condition (YPD), and after filtering out enzymes with no effect and isoenzymes, we identified 14 enzymes that significantly modify the response to mutations (Fig 2, S1 Table, Methods). Within this set, we also recognized five cases of particularly strong

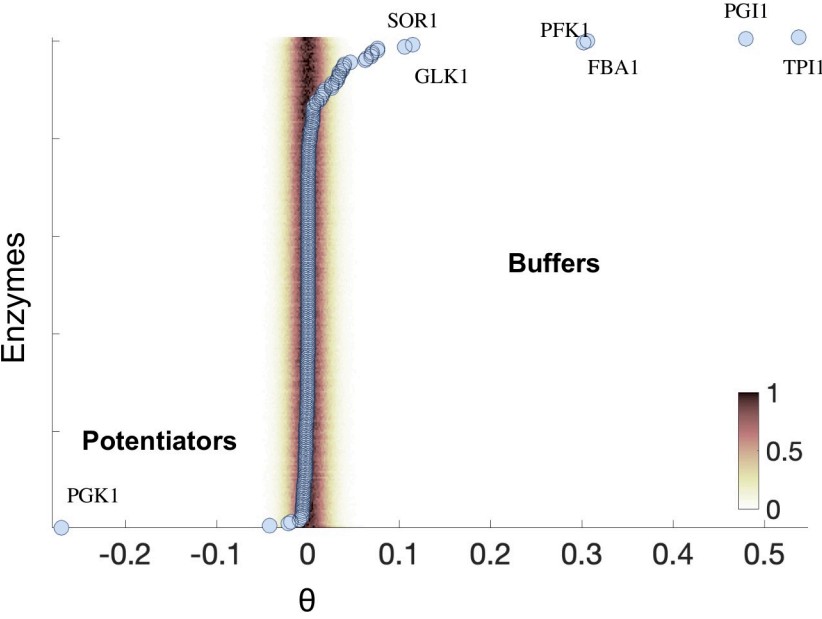

**Fig 2. Buffers and Potentiators in yeast metabolism.** For each enzyme, a θ score is computed, which is proportional to the difference in variability between the mutant and wild-type backgrounds. The shadow denotes the normalized null probability distribution of getting a particular score for each metabolic background. We added the names of the most significant modifiers (buffers with θ>0 and potentiators with θ<0).

effects, which are all related to the glycolysis/gluconeogenesis pathway: PGK1 (potentiator) and TPI1, PGI1, FBA1 and PFK1 (buffers). Deletion of these enzymes leads to particularly strong flux rewiring (observed rewiring = 76% of the total flux in the wild-type, mean rewiring expected randomly = 2%, random permutation test, p < 1e-4, with 10000 permutations) low fitness of the associated mutated metabolism (observed relative fitness by FBA = 0.2, mean relative fitness expected randomly = 0.98, random permutation test as before, p < 1e-4) and a more extensive number of MA lines with no growth (observed number of lethal MA lines = 1016, mean number of lethal lines expected randomly = 36, random permutation test as before, p < 1e-4; total number of lines = 10000); all features denoting the occurrence of very strong metabolic readjustments due to the enzyme deletion (Methods).

## Metabolic rationale underlying buffers and potentiators

The advantage of *in silico* models is that one can uncover these readjustments. Thus, an enzyme works as a potentiator when its absence frequently disables the costs of mutating a significant number of reactions, included in the MA lines, which decreases variability in growth rate ($std_{mutant} < std_{wild-type}$). To evaluate this, we identified those reactions enriched in MA lines whose impact on growth rate decreased in the ΔPGK1 background. The top five belong to the glycolysis-gluconeogenesis system and pyruvate metabolism. This is reasonable considering that PGK1 (3-PhosphoGlycerate Kinase) is a central enzyme whose mutation inactivates the fluxes on these pathways. The cost in growth of a mutation on these reactions is, therefore, smaller than in the wild-type background.

Enzymes working as buffers have the opposing effect. In this case, the absence of a buffer amplifies the weight of a substantial number of mutations found in the MA lines, increasing the variability in growth ($std_{mutant} > std_{wild-type}$). Which type of mutations show this amplification depends again on the effect of the specific background. If we first consider the top four buffers with a strong effect, we identify several reactions that considerably increased the flux in the corresponding metabolic background, like those related to alternative carbon metabolisms, e.g., glycerol, sorbitol, etc. Note also that a difference in flux variability relates to the explanation of when an enzyme works as a buffer or potentiator (Methods).

Beyond the specifics of the metabolic readjustments, both epistasis and pleiotropy have been argued to be relevant features to interpret buffers and potentiators. They quantify the number of interactions, with other mutations, and the functional role of these elements, respectively [12]. We consequently examined both features by computing the epistatic network between every pair of enzymes [31] (but note that higher-order interactions are also important [32, 33]) and a recently introduced metabolic pleiotropic score that quantifies the contribution of an enzyme to every biomass precursor [34, 35]. Global modifiers show strong pleiotropy and epistasis (S1 Table). This indicates overall their multifunctionality character, as illustrated in Fig 3, which shows pleiotropy and the number of weak negative genetic interactions. This type of genetic interaction appears when there exists an additional less efficient metabolic solution to the two main functional alternatives represented by the interacting genes. The multiplicity of alternatives with different efficiency usually reflects the presence of (qualitatively) different ways to perform a specific function [36] (Methods).

## Buffers and potentiators are condition dependent

These results confirm the intrinsic presence of buffering and potentiation elements modulating the response to mutations in biological networks, a result discussed before only with the use of simple gene-regulatory network models and that we extend here to a representative metabolic setting. Moreover, and given that the function of metabolic networks strongly depends

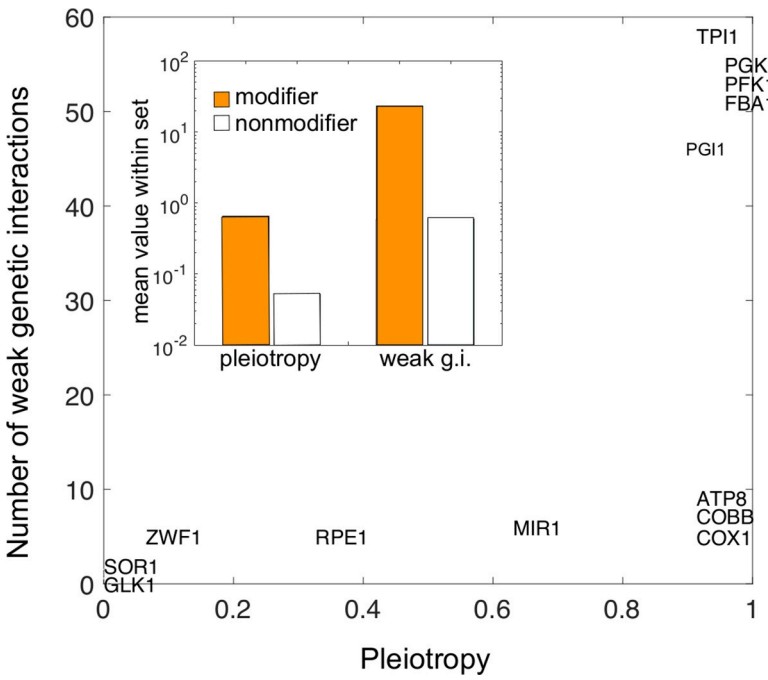

**Fig 3. Buffers and potentiators correspond to multifunctional enzymes.** We computed the pleiotropy and number of weak negative genetic interactions as proxies of enzyme multifunctionality (see main text and Methods). Modifiers (both buffers and potentiator) show stronger pleiotropy (mean pleiotropy modifiers = 0.65, mean pleiotropy nonmodifiers = 0.05, two-sample Kolmogorov-Smirnov test p = 1.6e-6) and number of weak negative genetic interactions (g.i.) (mean number of weak g.i. modifiers = 23.38, weak g.i. nonmodifiers = 0.6, KS p = 6.9e-8).

on the precise growing conditions [37], we could expect that most enzymes modify the response to mutations in a condition-dependent manner. To examine this, we studied two complementary situations (Methods). One in which we modify the carbon source complementing YPD conditions (that includes glucose by default), and a second one in which we studied a range of random nutrient conditions (from poor to rich media), and also minimal medium.

As projected, the list of enzymes acting as buffers or potentiators generally changes, with some enzymes precisely related to the specific growing conditions (S2 Table). For instance, GAL1, GAL7, and GAL10 (related to galactose metabolism) act as potentiators in YPG medium (galactose as carbon source), while glycerol utilization enzymes (GUT1 and GUT2) are potentiators in YPGly (glycerol as carbon source). Other enzymes switch their role, e.g., TPI1 (Triose-Phosphate Isomerase) functions as a buffer when growing in minimal medium or a potentiator in YPGly. In contrast, COX1, COBB, and ATP8 consistently buffer variation. Besides, and while there is a general tendency to exhibit more buffering than potentiation, there exist situations in which potentiation is dominant and others in which the number of enzymes acting as buffers is severely reduced. This emphasizes that the role of a particular enzyme in modifying the impact of mutations is a systemic feature of metabolism that depends on its regime of activity, i.e., alteration of environmental conditions matters.

## Are there enzymes acting as universal modifiers?

All the previous analyses distinguish a set of genes that can modify the amount of growth rate variability caused by the accumulation of mutations. We were also interested in studying to what extent these enzymes represent "universal" modifiers, i.e., their absence also changes the

response to other sources of phenotypic variation. If this were the case, it would suggest a single mode of canalization, i.e., the presence of broad mechanisms to alter the effect of perturbations [38]. Recent results argued though against this congruence [12, 39]. The debate is still open and surely depends on the level of the biological organization considered. We tried to examine this issue here with regards to two additional sources of variability; biochemical noise (related to the low copy number of molecules) and environmental fluctuations.

To generate the variability coupled to biochemical noise, we need first to compute the noise corresponding to the flux of each metabolic reaction in the network. We followed a previously established approach [40] (Methods), which uses data on expression noise of the enzymes (obtained in YPD medium) and explicit knowledge about the metabolic logic to subsequently estimate the reaction noise [40]. We can consider 10000 independent realizations where the flux of metabolic reactions is randomized depending on its noise (Methods). We thus obtain a distribution of growth rates for the wild-type metabolism and for those genetic backgrounds in which each of the enzymes is deleted. This permits us to compute a $\theta$ score as previously, but concerning the variability in growth rate due to biochemical noise: $\theta_{noise}$.

We noticed that the five strongest modifiers for mutations also appear as modifiers regarding noise; PGK1 as potentiator and PFK1, FBA1, TPI1, and PGI1 as buffers (although PGI1 emerges as a very strong buffer in this case instead of TPI1, the strongest buffer to variability caused by mutations). Three other "mutational" buffers remain as such: SOR1, RPE1, and GLT1, while new enzymes merely buffering variability due to noise also appear: GRE3, MAE1, CTT1, etc. (S1 Table).

We next examined the response to fluctuations in the environmental conditions [39]. By this, we mean deviations on the import fluxes that characterize YPD. To generate a fitness distribution, we computed growth rate in 10000 different environments in which the import of the corresponding nutrients fluctuates 10% of its fixed YPD value (Methods). Fitness distributions were computed for the wild-type and for all metabolisms in which one enzyme has been deleted to compute $\theta_{environment}$. This $\theta$ score is proportional to $std_{mutant}-std_{wild-type}$ as before.

In this setting, we find again that PGK1 acts as a potentiator and that PGI1, COBB, COX1, and ATP8 remain as buffers (S1 Table). Thus, two central enzymes act as a potentiator (PGK1) or buffer (PGI1) to all three sources of growth rate variability. Moreover, we computed the correlation of all three scores obtained (for every enzyme) as a measure of the similarity in the mode of canalization. We detect the strongest correlation between the mutational and the noise-induced variability (R = 0.77, p = 1.74e-101; mutational and environmental, R = 0.45, p = 6.58e-14, noise and environmental, R = 0.35, p = 1.76e-08).

## Discussion

The interconnectedness of biological systems, as revealed by the widespread identification of pleiotropic and epistatic effects [33], suggests that the presence of genetic modifiers of phenotypic variability should be a prevalent phenomenon [8, 11]. Simple gene-regulatory models [23] and morphometrics experiments in both yeast [41] and *Drosophila* [42] appear to confirm such a view. But finding additional cases to validate this principle is challenging given the insufficiency of large-scale experimental approaches to examine phenotypic variation with high resolution.

Here, we propose a complementary approach. We introduce the use of genome-scale metabolic models to generate large-scale quantitative phenotypic data. These models are not just simple toy models. They represent accurate representations of metabolism and also revealed as valid tools to provide predictions to be later confirmed experimentally, e.g., [43]. We show

that many enzymes work as buffers or potentiators of phenotypic variability originated by mutations in the reaction fluxes, with growth rate representing the complex phenotype.

This set is contingent on the precise working regime of the metabolism, e.g., the growing medium, emphasizing that this is an intrinsic property of the system generating the phenotype rather than of its constituents. In most of these regimes, we detected suppression of variation (buffering) as projected with simpler models [23], but there exist certain conditions in which potentiation predominates.

A particular enzyme might similarly be a modifier for other sources of variability [38]. We explicitly studied variability generated by the presence of biochemical noise or fluctuating environmental (nutrient) conditions. We find coinciding modifiers, i.e., congruence, between mutational and noise perturbations. However, given that our protocol to generate these two types of variability affects fluxes in a qualitatively similar manner, it is not surprising that we encounter similitude between the corresponding set of modifiers. Metabolism may neverthe-less represent a particular biological system where different perturbations eventually lead to the same response, but some disparities could be observed.

To better appreciate the rationale behind buffers and potentiators, we studied a specific condition (YPD). The most influential modifiers in this setting, which comprise the main enzymes of the glycolysis and respiratory chain, corroborate the significance of the multi-func-tionality of these elements within the network. Both sets of enzymes showed strong pleiotropy, which also correlates with the extent of metabolic rewiring and the amount of change of genetic interactions experienced when these enzymes are mutated [36]. That we uncover a similar set if we consider a different metabolic model (Methods) validates our exploration. More work on metabolic models would, of course, improve the individual predictions for a given condition [44].

Note that some of the most active modifiers mentioned before are enzymes of the glycolysis pathway (PGK1, FBA1, PGI1, PFK1, and TPI1) whose mutation considerably rewires metabo-lism (and consequently the impact of mutations on additional pathways). These enzymes also catalyze reactions needed for growth in non-sugar carbon sources, which can explain their repeated role as modifiers. Besides, four of them (except PFK1) are essential yeast genes for which the model also predicts substantial fitness costs (S1 Table). This hints to previous reports presenting essential genes as principal agents in regulating phenotypic variance [41]. The result was based on morphometric characterization of cells, so we chose to examine whether the modifiers we obtained here might represent modifiers to these additional traits. Variability is summarized in this case by introducing a phenotypic potential [both in nonessential [41] and essential genes [45], see Methods]: how much a mutation changes morphological variation. We plot the distribution of these scores in Fig 4, together with the precise value corresponding to the (metabolic) modifiers to fitness. Only two of them remain as modifiers.

Finally, our work has implications for the understanding of the consequences of phenotypic heterogeneity in tumors, emphasizing its very dynamic nature. Specific acquired mutations cause metabolic reprogramming (e.g., oncogenic drivers leading to characteristic metabolic sig-natures) that impacts growth. But the current knowledge of this reprogramming is somehow coarse; one mutation, or sequence of mutations, points to specific variations. Other recent work already hinted, however, to more context-specific results where the tissue of origin, or cell line-age, etc. [46, 47] alters the metabolic adjustments created by the same mutation. We have seen here an additional context effect. We showed how mutations influence the fitness effects of added mutations, and how they decisively shape the amount of heterogeneity in a population.

Furthermore, the tumor microenvironment could change the role of a precise gene muta-tion as buffer or potentiator of phenotypic variability [48], again as we have appreciated here with the dependence on nutrient conditions. These feedbacks eventually influence cancer

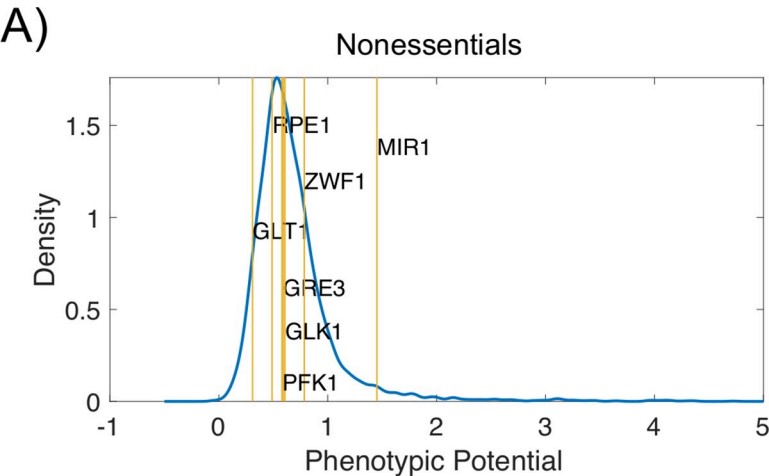

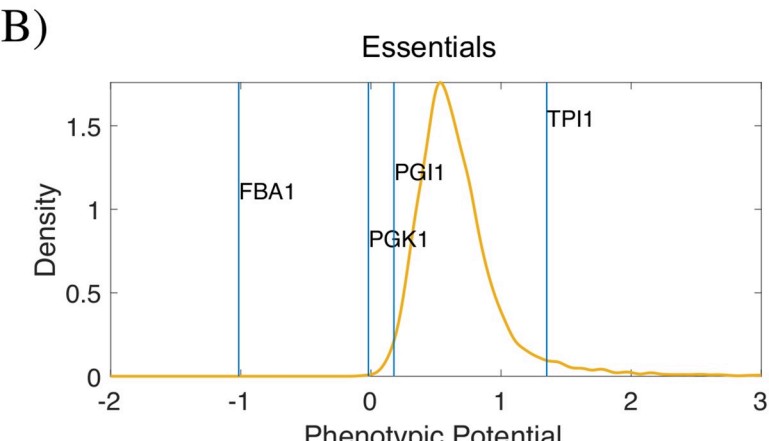

**Fig 4. Phenotypic potential to morphological variation.** The phenotypic potential scores the amount of morphological variation linked to a specific mutation (Methods). We show a kernel density plot of the distribution of scores for a collection of nonessential (A) and essential (B) genes, and the corresponding values for the set of modifiers to growth rate as phenotype (obtained in YPD; see Fig 2). MIR1 (nonessentials) and FBA1 (essentials) significantly exhibit a larger and smaller phenotypic potential than expected ($p < 0.05$, randomizing test), respectively.

progression and might have consequences in several therapies that are targeting different parts of metabolism, normally the glycolysis pathway. While the consequences of acting on certain targets might depend on the characteristic metabolic reprogramming linked to the genetic lesion and tissue type [49], this work accentuates that the outcome of metabolic inhibitors goes beyond the alteration of metabolism to the modification of the consequences in growth of subsequent mutations (see also S1 Note). Work is needed to assess the influence of this component in the application of effective interventions based on gene modifiers [50] to prevent disease.

# Materials and methods

## Models

We mostly worked with *Saccharomyces cerevisiae* iND750 [29] with a total number of 1266 reactions that incorporates all necessary complexity from yeast metabolism, while enabling us

to moderate the substantial computational load associated with our analysis. Standard conditions correspond to YPD rich medium (with 20 mmol gr$^{-1}$h$^{-1}$ of glucose and 2 mmol gr$^{-1}$h$^{-1}$ of $O_2$ import and an assortment of amino acids introduced at a rate of 0.5 mmol gr$^{-1}$h$^{-1}$). Reactions in the model are part of 56 subsystems linked to different metabolisms, e.g., fatty acids, glutamate, etc. Two of these subsystems correspond to exchange and biomass reactions (117 reactions) and bicarbonate ($HCO_3$) equilibration reactions. To validate the general appearance of buffers and potentiators in metabolism, we examined an additional *Saccharomyces cerevisiae* model (iAZ900 [51], in YPD medium). Using this model, we identified three potentiators (including PGK1), and sixteen buffers (including, ATP8, RPE1, SOR1, GLK1, COBB, COX1, TPI1, PGI1, FBA1, and PFK1) (S3 Table).

## FBA and MOMA

FBA is a mathematical tool for metabolic network analysis that allows the prediction of growth rate, i.e., fitness, and fluxes under the assumption of maximization of biomass production given a set of constraints. We use the Gurobi linear programming optimizer (www.gurobi.com) and the Cobra toolbox [52] in Matlab (www.mathworks.com). We also minimize the absolute value of fluxes to avoid loops in the solutions. We compute all reference metabolisms (wild-type and single-enzyme deletions, see below) with FBA. To obtain the fitness for each of the components of a MA mutation line, we used MOMA. A procedure that minimizes the deviation in fluxes from the corresponding metabolism without the mutations. MOMA outperforms the standard FBA approach in the prediction of growth rate and fluxes upon mutation. It relies on the assumption that after genetic perturbations, the organism's metabolic and regulatory responses favor a new steady state close to the original operating region, rather than maximizing cellular growth [30]. Note that both FBA and MOMA are extensively used to compute growth rate and mutant growth rates, respectively. These methods have some limitations, of course, which are always present to any modeling approach. E.g., the incorporation of gene regulation (this is part of current research in the flux balance community). However, none of these limitations make FBA or MOMA inadequate for our work.

## Generation of mutation accumulation lines

We produced 10000 independent "mutation accumulation lines" by fixing for each line the flux of 5% of the constituent biochemical reactions of the wild-type metabolism chosen at random (S1 Script, S4 Table). For each designated reaction, we assigned a random value obtained from a uniform distribution between 0 and 20 mmol gr$^{-1}$h$^{-1}$ to the corresponding lower (reversible reactions) and upper bounds of the associated flux. External exchange reactions (116 reactions) are not incorporated in the generation of the MA lines to maintain the nutrient conditions. One could explore, of course, other means to implement mutations, but this does not invalidate our approach and the results we found.

## Protocol to identify buffers and potentiators

Our goal is to quantify to what extent the accumulation of a fixed set of mutations ("MA lines") causes a different response in growth due to the presence or absence of a particular enzyme. We begin with a compilation of "reference" metabolisms that includes the wild-type and all possible variants with a single enzyme removed. The growth rate of these metabolisms is computed with FBA. After this, each reference metabolism experiences the very same set of mutations in the fluxes (the MA lines defined previously). For each line, fitness is calculated with MOMA with regards to deviations to the respective reference metabolism and normalized by the fitness value of the reference (all lines with the wild-type as a reference has nonzero

fitness). We calculated the variability on the (relative) fitness observed in the 10000 MA lines. If the variability observed in a specific mutant is bigger than that observed in the wild-type, we say that the corresponding enzyme is a buffer; if smaller, we say that it is a potentiator. We use the relative difference with respect to the wild-type value θ = (STD fit_mutant–STD fit_wild-type)/STD fit_wild-type [11, 22] as score. Different measures that we tested led to comparable results, like the genotype-by-line interaction variance [12] (S1 Table).

### Flux variability

Change of flux variability could also describe when an enzyme works as a buffer or potentiator. Flux variability maximizes and minimizes each flux of the metabolic network while satisfying all the other constraints that fix a given growth rate. This is equivalent to the experiments of the accumulation of mutations. Imagine those mutations constraining a particular flux. The associated fitness values represent a measure of how much flux variability can be tolerated. If one compares the very same mutations in a different genetic context (where a specific enzyme is deleted), one is quantifying again the flux variability, this time in that particular context.

### Flux rewiring

For each reference metabolism, we computed the Euclidean norm of the vector defined by the difference between the optimal FBA fluxes of the mutant background and the wild-type. We divided this value by the Euclidean norm of the optimal wild-type FBA flux. This measure indicates the degree of metabolic reprogramming experienced by a given mutant.

### Random environments, environments with a carbon source other than glucose and minimal medium

Random environments were aerobic (2 mmol $gr^{-1}h^{-1}$ of $O_2$ import; ammonia, phosphate, sulphate, sodium, potassium, $CO_2$, and $H_2O$ unbound), with the specific set of nutrients being selected from an exponential distribution probability [53] (with mean = 0.1). After defining this set, their dosage was randomly obtained by applying a uniform distribution between 0 and 20 mmol $gr^{-1}h^{-1}$ (S2 Table). We also examined some YPD variants, i.e., YPE, YPGal, YPGly, and YPLac, in which the import of glucose at 20 mmol $gr^{-1}h^{-1}$ as a carbon source is substituted by ethanol, galactose, glycerol, and lactate, respectively. Minimal medium provided unconstrained ammonium, phosphate, and sulphate with glucose import at 10 mmol $gr^{-1}h^{-1}$ and $O_2$ at 2 mmol $gr^{-1}h^{-1}$.

### Buffering-potentiation protocol regarding biochemical noise variability

We followed a procedure grounded on the one presented by Wang and Zhang [40] to simulate the noise in the flux of a reaction. Flux noise incorporates experimentally measured gene expression noise data [54] that largely excluded extrinsic noise (noise measured in YPD conditions) and approximates the metabolic network as a linear pathway of length $n$ (S1 Table). For a fixed $n$, we run 10000 simulations in which we constrain fluxes according to the noise and compute the corresponding fitness with MOMA (deviation to a noiseless metabolism) to obtain the variability associated with intrinsic noise (we presented $n = 4$ in the main text [40]). We apply this procedure for each reference metabolism (wild-type and mutants) so that we can define a θ score for the variability in growth rate associated with noise: $\theta_{noise}$.

## Buffering-potentiation protocol regarding environmental variability

We generated 10000 different environments by randomly modifying the bounds of the nutrient reactions defining the YPD medium while maintaining ammonia, phosphate, sulphate, sodium, potassium, $CO_2$, and $H_2O$ unbound and the import of $O_2$ to 2 mmol $gr^{-1}h^{-1}$. Fitness of the new environmental conditions was computed with MOMA with the corresponding metabolic solution in YPD as a reference and normalized by the fitness value of that reference metabolism. We computed the variability on this (relative) fitness to then define a $\theta$ score as before: $\theta_{environment}$.

## Pleiotropy and Epistasis

We applied FBA to compute the production rate of each biomass precursor for a given growth medium and genetic environment. To simulate the production of a given metabolite, we added a new exchange reaction to the model representing the secretion of this metabolite, and maximize the flux through this reaction [34, 35]. For the single-knockout annotation, we systematically deleted each gene and considered that it contributed to the production of a certain metabolite if its loss reduced the metabolite's production rate by more than 20%. We divided the number of metabolites for which a gene contributes by the total number to obtain a normalized score between 0 and 1. This number describes the multifunctionality at the network level, and hence, the pleiotropy of the corresponding gene [34, 35]. To compute the epistatic network, we calculated with FBA the growth rates of all single and double deletion mutants encompassing all nonessential genes. The mutant/ WT growth ratios obtained are used to compute an epistatic score ($\varepsilon$), which incorporates a multiplicative model and posterior scaling [31, 36]; interactions with $|\varepsilon| < 0.01$ were not considered.

## Phenotypic potential

Morphological phenotypes of individual cells are available for two sets of knockout strains of nonessential [41] and essential [45] genes, in which a single measure of phenotypic variance–termed the phenotypic potential–was obtained. Note, however, that this measure is not completely equivalent in the two sets.

## Supporting information

**S1 Note. Genetic modifiers and expression variability in cancer.**
(PDF)

**S1 Fig. Mutations characterizing specific tumors increase the enzyme expression variability as compared to normal tissues.** We used gene expression data of pairs of control and tumor samples to quantify the variability (standard deviation) in the expression of metabolic genes within each sample (see S1 Note for details). With these scores, we estimated the fraction of enzymes with more variation within the tumor sample than the control (this ratio is indicated by the orange bars, in increasing order). We also computed the expected null value of this score by randomization of expression data between tissue and control. We plot the mean null value of these randomizations (blue curve) and the +/- 2 std (blue shading, 1000 randomizations).
(TIF)

**S1 Table.** List of enzymes features including 1/ $\theta$ scores of changed fitness variability associated with mutations, noise and environment, 2/ variances of wild-type, mutant and interaction, and 3/ number of lines which are lethal, 4/ pleiotropy, number of genetic interactions,

flux rewiring and FBA growth rates.
(XLSX)

**S2 Table. List of enzymes acting as buffers and potentiators in different media.**
(XLSX)

**S3 Table. Buffer and potentiator study for the iAZ900 metabolic model of *S. cerevisiae*.**
(XLSX)

**S4 Table. List of mutations experienced in the mutation accumulation lines.**
(XLSX)

**S1 Script. Core Matlab script describing how simulations are done, with all settings of flux bounds included.**
(PDF)

## Acknowledgments

I would like to thank Eugene Plavskin and Mark L Siegal for discussions and the Center for Genomics and Systems Biology, New York University, for their hospitality while this research was conducted. I also thank Alvar Alonso-Lavin, Djordje Bajić, Mónica Chagoyen, and Pablo Yubero for comments on an earlier version of the manuscript.

## Author Contributions

**Conceptualization:** Juan F. Poyatos.

**Investigation:** Juan F. Poyatos.

**Resources:** Juan F. Poyatos.

**Writing – original draft:** Juan F. Poyatos.

**Writing – review & editing:** Juan F. Poyatos.

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
