## [Decision Letter · Decision Letter 0]

15 Apr 2020

Dear Dr. Poyatos,

Thank you very much for submitting your manuscript "Genetic buffering and potentiation in metabolism" for consideration at PLOS Computational Biology.

As with all papers reviewed by the journal, your manuscript was reviewed by members of the editorial board and by several independent reviewers. In light of the reviews (below this email), we would like to invite the resubmission of a significantly-revised version that takes into account the reviewers' comments.

Two reviewer and I read the paper carefully. I think it is fair to say that we agree:

-- the rationale/"story" seems reasonable and interesting;

-- and the numerical experiment also seems reasonable.

However, the big concern is: not every reasonable story turns out to be true. That is, there is a serious lack of support to the findings.

Obviously, the best will be to have support from experimental data. Or at least use EXTENSIVE literature findings and bioinformatics

analysis to support your findings.

I would like to add a note here which may assist you make decisions in revision: if there is no independent experimental data to support,

it is hard to predict whether the reviewers and I will find the evidence sufficiently convincing.

A major/minor point is: some parts of the paper are not very "friendly". They are too vague and too broad. Overall please try to make

your discussions more concrete.

We cannot make any decision about publication until we have seen the revised manuscript and your response to the reviewers' comments. Your revised manuscript is also likely to be sent to reviewers for further evaluation.

Sincerely,

Shuangge Ma

Guest Editor

PLOS Computational Biology

Daniel Beard

Deputy Editor

PLOS Computational Biology

Two reviewer and I read the paper carefully. I think it is fair to say that we agree:

-- the rationale/"story" seems reasonable and interesting;

-- and the numerical experiment also seems reasonable.

However, the big concern is: not every reasonable story turns out to be true. That is, there is a serious lack of support to the findings.

Obviously, the best will be to have support from experimental data. Or at least use EXTENSIVE literature findings and bioinformatics

analysis to support your findings.

I would like to add a note here which may assist you make decisions in revision: if there is no independent experimental data to support,

it is hard to predict whether the reviewers and I will find the evidence sufficiently convincing.

A major/minor point is: some parts of the paper are not very "friendly". They are too vague and too broad. Overall please try to make

your discussions more concrete.

Reviewer's Responses to Questions

**Comments to the Authors:**

Reviewer #1: Minor Grammatical errors

If the authors can validate the claims by existing biological (wet lab data ) in this study it would be better. As there seems to be no validation.

As even though never done before you would expect the Glycolysis pathway and the enzymes associated with it would have a significant impact

Reviewer #2: The author used Saccharomyces cerevisiae metabolic model to examine genetic modifiers that serving as buffers or potentiators of variability. The perspective and experiment design are quite interesting and novel. Systemic evidence of evolutionary capacitor or potentiator act as a common feature in biological networks is rare. Although this study is an in-silico exploration, it could provide rational insight into this issue. The manuscript needs may need some modification.

1, The author may want to introduce more on what the metabolic model is and how it fits in the simulation of evolutionary studies.

2, Is there any reason why to choose iND750 model as the main model for simulation instead of a more recent metabolic model which contains more reactions and genes?

3, The author may want to specify the setting of lower and upper bound of flux for those unlimited reactions compare to the limited flux. And please provide the Matlab code in how the simulation is done that can be posted on the journal website as supplementary material to increase the impact of the paper.

4, In the results part the author discussed the potential buffers and potentiators that was identified and its properties. Is there any corroborating experimental evidence shows that these enzymes working as buffers or potentiators? Is it possible to verify the findings?

5, The author mentioned that FBA and MOMA have different objective functions. The model implements enzyme knockout by constrain all the flux of the reactions that the enzyme involved in the model to zero which is a similar approach the author construct MA lines. Wouldn’t it be more consistency to use only FBA or MOMA to calculate the variability of growth rate in different simulations?

6, The discussion on how this work can contribute to cancer study seems conceptual and theoretical. The author may want to make changes on the last two paragraphs to make it easy to follow.

7, Page 8 line 247 -250, The sentence “Besides, and while there is a general tendency to exhibit more buffering that potentiation, there exist situations in which potentiation is dominant and others in which the number of enzymes acting as buffers is severely reduced.” seems confusing. Does the author mean “Besides, and while there is a general tendency to exhibit more buffering than potentiation, …”?

8, Page 17 line 541, there is two “e.g.” In the sentence. The author may want to carefully review the manuscript to avoid these minor mistakes.

**Have all data underlying the figures and results presented in the manuscript been provided?**

Reviewer #1: No: 1000 gene mutation data should be provided

Reviewer #2: Yes

PLOS authors have the option to publish the peer review history of their article (what does this mean?). If published, this will include your full peer review and any attached files.

Reviewer #1: No

Reviewer #2: No
---

## [Decision Letter · Decision Letter 1]

22 Jun 2020

Dear Dr. Poyatos,

Thank you very much for submitting your manuscript "Genetic buffering and potentiation in metabolism" for consideration at PLOS Computational Biology.

As with all papers reviewed by the journal, your manuscript was reviewed by members of the editorial board and by several independent reviewers. In light of the reviews (below this email), we would like to invite the resubmission of a significantly-revised version that takes into account the reviewers' comments.

As you may see, one reviewer was satisfied with your revision. But the other reviewer actually raised his/her level of concern.

Please do address carefully.

I agree with the comment on the (lack of) link with cancer. But this can be relatively easily solved.

I also agree with the reviewer that the lack of validation/justification is a major problem. A paper cannot be published simply on the "trust me" ground.

Thanks for your attention.

We cannot make any decision about publication until we have seen the revised manuscript and your response to the reviewers' comments. Your revised manuscript is also likely to be sent to reviewers for further evaluation.

Sincerely,

Shuangge Ma

Guest Editor

PLOS Computational Biology

Daniel Beard

Deputy Editor

PLOS Computational Biology

As you may see, one reviewer was satisfied with your revision. But the other reviewer actually raised his/her level of concern.

Please do address carefully.

I agree with the comment on the (lack of) link with cancer. But this can be relatively easily solved.

I also agree with the reviewer that the lack of validation/justification is a major problem. A paper cannot be published simply on the "trust me" ground.

Thanks for your attention.

Reviewer's Responses to Questions

**Comments to the Authors:**

Reviewer #1: The lists for all 10000 mutations need to be provided as supplementary files.

The corroboration of the study with cancer is still not completely elaborated. The work is linked to cancer just in the discussion, it js a well studied phenomenon that driver mutations propagate other mutations and all this depends on the type of cancer, onset, age and many other factors. It is felt that the work is forced linked to cancer. There is no link to the environments mentioned, to the genes mentioned just that tumor micro environment. The Supplementary note 1 is inadequate. It is already tumour environments are heterogeneous and hence there is going to gene expression variability, hence the enzymes, variability. What is the link to your work is not well established.

If the researchers cannot do the mutational analysis to validate their claims they need to find existing research works that help in validation. Line 160 - 168 it is mentioned that 100 such lines (ref 21) have been generated, do the results of these 100 match your work? Does your work complement the ref stated?

The discussion needs to be modified to explain ans elaborate more on the genes mentioned in the results as in why do the researchers think these are universal (PGK1 as potentiator and PFK1, FBA1, TPI1 and PGI1 - line 283). Giving biological reasoning rather than just mentioning they are part of important metabolic pathway as they have done.

Reviewer #2: The manuscript has made significant improvement and is now suitable for publication.

**Have all data underlying the figures and results presented in the manuscript been provided?**

Reviewer #1: No: 10000 mutations list and all the data generated from it needs to be made accessible

Reviewer #2: Yes

PLOS authors have the option to publish the peer review history of their article (what does this mean?). If published, this will include your full peer review and any attached files.

Reviewer #1: No

Reviewer #2: No
---

## [Decision Letter · Decision Letter 2]

23 Jul 2020

Dear Dr. Poyatos,

We are pleased to inform you that your manuscript 'Genetic buffering and potentiation in metabolism' has been provisionally accepted for publication in PLOS Computational Biology.

Best regards,

Daniel A Beard

Deputy Editor

PLOS Computational Biology

Daniel Beard

Deputy Editor

PLOS Computational Biology

thanks for addressing all the concerns. congratulations on a work nicely done.

Reviewer's Responses to Questions

**Comments to the Authors:**

Reviewer #1: The authors have responded adequately

**Have all data underlying the figures and results presented in the manuscript been provided?**

Reviewer #1: None

PLOS authors have the option to publish the peer review history of their article (what does this mean?). If published, this will include your full peer review and any attached files.

Reviewer #1: No

---

## [Editor Report · Acceptance letter]

26 Aug 2020

PCOMPBIOL-D-20-00378R2 

Genetic buffering and potentiation in metabolism

Dear Dr Poyatos,

I am pleased to inform you that your manuscript has been formally accepted for publication in PLOS Computational Biology. Your manuscript is now with our production department and you will be notified of the publication date in due course.

With kind regards,

Matt Lyles
